# Host Factors Promoting the LTR Retrotransposon Life Cycle in Plant Cells: Current Knowledge and Future Directions

**DOI:** 10.3390/ijms27010374

**Published:** 2025-12-29

**Authors:** Pavel Merkulov, Alexander Polkhovskiy, Elizaveta Kamarauli, Kirill Tiurin, Alexander Soloviev, Ilya Kirov

**Affiliations:** 1Moscow Center for Advanced Studies, Kulakova Str. 20, 123592 Moscow, Russia; paulmerkulov97@gmail.com (P.M.); polkhovsky.a.w@gmail.com (A.P.); kamarauli2001@yandex.ru (E.K.); tiurin.kn@gmail.com (K.T.); 2All-Russia Research Institute of Agricultural Biotechnology, Timiryazevskaya Str. 42, 127550 Moscow, Russia; 3All-Russia Center for Plant Quarantine, Pogranichnaya Str. 32, 140150 Ramenskoe, Moscow Region, Russia

**Keywords:** LTR retrotransposons, genome evolution, plant genomes, mobilome, host factors, retrotransposon life cycle, retrotransposition, multiomics, interactomics, reporter systems

## Abstract

Long Terminal Repeat (LTR) retrotransposons (LTR-RTEs) comprise up to 90% of some plant genomes and drive genome diversification through their amplification. Novel insertions arise during the final stages of the LTR-RTE life cycle, which depends on both LTR-RTE-encoded proteins and host cellular factors. The LTR-RTE elements require host transcriptional machinery for RNA production, followed by nuclear processing/export, translation, virus-like particle assembly, reverse transcription, and genomic integration. This review addresses the following question: What host proteins promote LTR-RTE transposition in plants? Our analysis of recent literature on host factors and cellular compartments implicated in the retrotransposition cycle reveals the extensive integration of LTR-RTEs into host processes. Nonetheless, the precise mechanisms remain poorly resolved, especially in plants with their rich repertoire of LTR-RTEs. We propose integrating plant mobilomics with transposition reporters, genome editing, synthetic biology, and interactomics to elucidate plant-specific mechanisms.

## 1. Introduction

Long Terminal Repeat (LTR) retrotransposons (LTR-RTEs) represent a dominant yet enigmatic component of plant genomes. They often constitute the largest portion of plant genomes, driving plant evolution and genome size variation. The ongoing insertion activity of LTR-RTEs contributes to genomic diversity by creating structural variants, influencing gene expression, and generating phenotypic diversity critical for adaptation and evolution [1]. These elements can insert near or within genes, modulating their activity via epigenetic mechanisms such as RNA-directed DNA methylation, which impacts plant development and stress responses. In plant breeding, LTR-RTEs hold potential as tools for creating genetic diversity and novel traits, as spontaneous or induced retrotransposition events can generate new alleles that breeders can select [1,2,3]. Although ubiquitous, the life cycle of LTR-RTEs and their interactions with host factors and cellular compartments remain poorly characterized. LTR-RTEs have a life cycle that is similar to retroviruses and includes several steps: transcription from their genomic loci, RNA export from the nucleus, translation in the cytoplasm, assembly of virus-like particles (VLPs), within which the RNA is reverse transcribed. The newly synthesized complementary DNA (cDNA) is imported back into the nucleus and integrated as a new copy into the host genome. A growing number of studies on yeast, fruit fly, and vertebrates have reported that the life cycle of LTR-RTEs exhibits complex interactions with host factors [4,5]. For example, genome-wide mutational screening and transposition assays of *Tf1* retrotransposon in *Schizosaccharomyces pombe* revealed that *Tf1* requires an extensive network of host factors for successful integration. Genome-wide screens have identified 61 genes that promote *Tf1* integration, including factors involved in transcription, nuclear transport, mRNA processing, vesicle transport, chromatin structure, and DNA repair [4]. Moreover, two of these proteins, Rhp18 and the NineTeen complex, were confirmed to interact directly with *Tf1* integrase. A recent study identified more than 100 host proteins whose knockdown significantly reduced retrotransposon abundance in *Drosophila*, highlighting key regulators of LTR-RTEs [5]. In plants, two genome-wide association (GWAS) studies showed that tens to hundreds of genes are associated with TE copy number variation and TE expression in *Arabidopsis thaliana* [6]. While this data requires experimental verification, it shows that the plant LTR-RTE—host interaction network is very complex. Importantly, the findings in yeast draw parallels to *Drosophila* transposon regulation, where several homologous DNA repair components, including nucleotide excision repair factors (e.g., *RAD3*, *RAD25*) and the alternative end-joining (alt-EJ) DNA repair proteins (*Fen1* and *XRCC1*), modulate Ty1 retrotransposon activity [7,8]. The overlap in gene ontologies and regulatory factors between yeast and *Drosophila* retrotransposon screens underscores the conserved mechanisms by which host DNA repair-related proteins influence retrotransposon propagation across eukaryotes [5].

This review explores the current knowledge of host proteins that facilitate LTR-RTE transposition in plants. The central question of this review is: “What host cellular factors and mechanisms promote the LTR-RTE life cycle in plants?”. To address this, we summarize existing insights into host factors supporting retrotransposition across diverse organisms, from yeast to mammals, and highlight the present state of understanding in plants. We systematically examine each stage of the retrotransposition cycle—transcription, RNA processing, translation, reverse transcription, nuclear import, and integration—outlining the molecular and cellular mechanisms identified to date. Our analysis emphasizes that the life cycle of LTR-RTEs is deeply integrated within the host cellular environment and depends on numerous host proteins. However, the molecular mechanisms underlying these interactions remain poorly understood, and information specific to plants as organisms that harbor the most diverse repertoire of LTR-RTEs is particularly limited. To bridge this knowledge gap, we propose combining plant mobilomics with reporter-based transposition systems and leveraging advanced methodologies in genome editing, synthetic biology, and interactomics.

## 2. LTR-RTE Transcription, RNA Export, and Modifications

### 2.1. LTR-RTE Transcription

The first step in the LTR-RTE life cycle is transcription, which initiates from promoters embedded in the 5′ LTR and is regulated by host factors across diverse eukaryotes [9]. The abundance and expression patterns of LTR-RTEs vary considerably across species and tissues, often demonstrating transcriptional activation through recognition of LTRs by specific transcription factors and host protein complexes. LTRs frequently contain cis-regulatory motifs that recruit host transcription factors, allowing these elements to exploit developmental and stress-related signaling pathways [10]. A well-studied example is the heat-inducible *ATCOPIA78*/*ONSEN* family in *Arabidopsis thaliana*. Under heat stress, the transcription factor HSFA1 activates HSFA2, which then binds directly to a heat-responsive element (HRE) upstream of the *ONSEN* transcription start site, thereby inducing strong transcriptional activation [11]. Beyond heat shock factors, LTR-RTEs interact with diverse stress-responsive transcription factor families. Transcription of *EVADE*/*ATCOPIA93* in *Arabidopsis* is activated by an immune signal through direct binding of WRKY transcription factors to W-box elements in the LTR. This induction is possible only upon weakening of epigenetic repressive barriers such as DNA methylation and H3K27m3 modification [12]. In moso bamboo (*Phyllostachys edulis*), *PHRE1* and *PHRE2* LTR-RTEs interact with TCP20, DOF2, and GATA transcription factors [13]. In *Saccharomyces cerevisiae*, Ty1 retrotransposon transcription is tightly controlled by host chromatin remodeling complexes. For example, the SWI/SNF chromatin remodeling complex, comprising the core subunits Snf2, Snf5, and Snf6, has been shown to be essential for efficient Ty1 transcription [14]. Conversely, the *Fun30* family remodelers Fft2 and Fft3 cooperatively maintain high nucleosome occupancy at LTR elements, enforcing the use of downstream transcription start sites that produce truncated, non-functional RNA molecules. Under stress conditions, *Fun30* proteins are downregulated, leading to reduced LTR nucleosome occupancy and a shift to productive transcription start sites. The stress-responsive regulation of Fun30-mediated nucleosome positioning provides a sophisticated regulatory switch that allows rapid stress-induced LTR-RTE activation while maintaining tight control under normal conditions. Plants also possess Snf2-like chromatin remodelers, such as DDM1, which maintains DNA and histone methylation in heterochromatic regions by regulating nucleosome occupancy and composition, thereby influencing LTR-RTE transcription [15]. Similarly, the INO80 chromatin remodeling complex, recently characterized in rice (OsINO80), promotes H3K27me3 and H3K9me2 deposition to maintain LTR-RTE silencing [16]. Unlike yeast Fun30 remodelers that control transcription start site selection plant mechanisms primarily involve nucleosome composition changes, DNA methylation and histone modifications rather than TSS control. Thus, chromatin remodeling complexes regulate LTR-RTEs across eukaryotes, but in plants they primarily function in silencing rather than promoting transposition.

Like host protein-coding mRNAs, transcripts of most LTR-RTEs are synthesized by RNA polymerase II and typically carry a 5′ cap and a poly(A) tail (Figure 1) [9,17,18,19,20,21,22]. However, an exception is found in *Cassandra* retrotransposons of the TRIM (terminal-repeat retrotransposons in miniature) family, which contain conserved regions homologous to 5S rRNA within their LTRs and are transcribed by RNA polymerase III, generating uncapped and non-polyadenylated RNAs (Figure 1) [23]. In *Drosophila*, additional host mechanisms modulate LTR-RTE expression. Recent discoveries have revealed that certain DNA repair proteins can facilitate rather than restrict retrotransposon transcription, challenging traditional assumptions about the relationship between DNA repair machinery and transposable element control [5]. Translesion DNA polymerases Rev1 and PolH promote retrotransposon transcription by maintaining RNA polymerase II occupancy at retrotransposon loci (Figure 1). This activity is particularly critical at R-loop–rich regions, where these polymerases help to bypass transcriptional obstacles, enabling efficient expression of elements such as *3S18* and *HMS-Beagle* and ensuring sufficient retrotransposon RNA accumulation for mobilization [5].

### 2.2. Nuclear Export of LTR-RTE RNAs

Following transcription, the next critical step is the nuclear export of LTR-RTE RNA to the cytoplasm for translation and particle formation. Although direct studies of LTR-RTE RNA export in plants have not been conducted, the conservation of individual components of the mRNA export machinery suggests similarities to mechanisms operating in more well-studied systems [24,25].

Plant mRNA nuclear export involves several protein complexes that function differently depending on whether mRNAs contain introns or are intronless. It is worth noting that plant LTR-RTEs generally lack introns, with notable exceptions including *Ogre* elements in *Medicago truncatula* and the *EVADE Ty1*/*Copia* element in *Arabidopsis*, which contain functional introns important for generating subgenomic RNAs through alternative splicing [26,27]. Therefore, LTR-RTEs require mechanisms for both splicing-dependent export (for intron-containing variants like *EVADE* and *Ogre*) and splicing-independent mRNA export (for intronless elements).

The TREX (transcription and export) complex is a central component of plant mRNA export machinery [24,25]. The plant TREX complex includes the THO subcomplex, the TEX1 protein (a functional homolog of yeast *Tex1* and a core component of THO), ALY family adaptors (orthologs of yeast *Yra1* and human *ALYREF*), and *MOS11*, an ortholog of yeast *Tho1* and human *CIP29* [24,28,29]. Unlike yeast and mammalian systems that employ the TAP/NXF1 export receptor, plants utilize the TREX complex and associated factors as their primary mRNA export machinery [24,25].

For intronless mRNA export in plants, the TREX complex components can be recruited to mRNAs via cap-binding complex (CBC) interaction [24]. The CBC associates co-transcriptionally with the 5′ cap structure and facilitates recruitment of ALY and other TREX components to nascent mRNAs [30]. Alternative mechanisms for intronless mRNA export have also been described, including recognition of specific cis-elements called cytoplasmic accumulation regions (CARs) by TREX complex components [31]. These CARs are relatively large elements (160–285 nt) that contain one or more 10-nt consensus sequence motifs within them, and these elements directly recruit the TREX export machinery [31,32]. Thus, spliced LTR-RTE transcripts in plants can recruit export machinery through CBC interaction and potentially through additional mechanisms involving splicing factors, while intronless LTR-RTE genes rely primarily on CBC-mediated and CAR-element-based recruitment of TREX components for mRNA export [24,25,27]. Whether plant LTR-RTEs possess CAR-like motifs remains unclear. Instead of this, plant LTR-RTE pararetroviruses (e.g., Cauliflower mosaic virus) export their polycistronic mRNAs through a distinct mechanism involving a highly structured 5′ leader region (approximately 482–608 nt) that recruits TREX complex components, particularly MOS11, in addition to viral proteins P4 and P5 that serve as export adaptors [24].

In mammals, the principal mRNA export receptor NXF1 can be recruited to retroviral and retrotransposon RNAs either directly via specific cis-acting RNA structural elements or indirectly via the TREX adaptor complex [33,34,35]. For example, the mouse LTR-RTE *IAP* and *musD* contain specialized cis-RNA elements, RTE (RNA transport element) and *MTE* (*MusD* transport element), that mediate splicing-independent nuclear export of full-length transcripts via the NXF1/NXT1 pathway: RTE directly recruits the cofactor RBM15, whie MTE forms a complex intramolecular RNA structure (with pseudoknot and kissing loop motifs) that facilitates export (Figure 1). These elements allow retrotransposon RNA to bypass the usual dependence on splicing-dependent TREX recruitment and efficiently enter the cytoplasm for translation [35,36]. In *Caenorhabditis elegans*, a unique nuclear protein encoded by spliced LTR-RTE *Cer1* mRNA was discovered that is required for the export of unspliced *Cer1* RNA from the nucleus to the cytoplasm and can form giant nuclear structures (Figure 1) [37].

Proteins encoded by individual LTR-RTEs have been shown to be involved in the export of their mRNA. For *Ty1* elements of *S. cerevisiae*, the involvement of GAG protein in mRNA export from the nucleus has been demonstrated. Following initial GAG-independent export of full-length LTR-RTE RNA (gRNA) and its translation, the synthesized GAG associates with the nuclear envelope, where it promotes further export and stabilization of mRNA and the formation of cytoplasmic GAG-mRNA foci [38]. In *S. cerevisiae*, specific nuclear pore complex (NPC) components modulate *Ty1* at multiple stages: Nup60 and Nup159 are required for optimal *Ty1* mRNA and GAG levels (Figure 1), while other basket and inner-ring Nups affect mobility and integration targeting, indicating NPC-mediated regulation that extends beyond a single mRNA export pathway [39].

### 2.3. Processing and Modification of LTR-RTE RNAs

The accumulated evidence demonstrates that RNA post-transcriptional processing and modifications represent critical and indispensable steps in the LTR-RTE life cycle. These processes are facilitated by host-encoded proteins that are highly conserved across diverse organisms. One of the earliest LTR-RTE RNA modifications occurring during transcription is capping, which adds a 7-methylguanosine (m^7^G) cap to the 5′ end of nascent pre-mRNA. This modification enables efficient nuclear export, stability against exonucleases, and translation initiation for functional proteins, while avoiding packaging into (VLPs) [40]. Unlike standard mRNA biogenesis where capping promotes translation, both capping and decapping regulate distinct LTR-RTE RNA pools: capped transcripts support protein translation, whereas uncapped or decapped ones are directed to replication. In *S. cerevisiae*, *Ty1* RNA is colocalized with 5′-3′ decay factors including Dcp1, Dcp2, and Xrn1 (Figure 1). Loss of *Xrn1* abolishes *Ty1* RNA packaging into VLPs, indicating that 5′-3′ mRNA decay is required for creating a packageable, non-translating RNA pool, in addition to its canonical roles in mRNA turnover [40]. For plant LTR-RTEs, direct evidence for post-transcriptional decapping to generate packaging substrates is currently lacking, but the barley *Ty1*/*Copia* element *BARE* employs a dual-promoter strategy that yields distinct RNA classes with different fates. Specifically, TATA2-derived RNAs are capped, polyadenylated, and polyribosome-associated for translation, whereas TATA1-derived RNAs are uncapped and non-polyadenylated, and serve as packaging substrates with terminal repeats required for reverse transcription and VLP replication [41,42] (Figure 1). In addition to the formation of full-length and sense transcripts, decapping is also required for reducing the amount of antisense LTR-RTE transcripts that are frequently formed by many LTR-RTEs [43]. Their presence can lead to the formation of dsRNA, which acts as a trigger for silencing mechanisms, or reduce retrotransposition efficiency, potentially by interfering with reverse transcription within VLPs [44,45]. Studies of the *Ty1* antisense regulatory RNA (RTL) in *S. cerevisiae* show that decapping by the Dcp1/Dcp2 proteins and 5′-3′ decay by the exoribonuclease Xrn1 eliminate RTL and thereby antagonize its trans-silencing activity [46]. Notably, *Xrn1* orthologs from multiple *Saccharomyces* species, when expressed in *S. cerevisiae*, equivalently support *Ty1* retrotransposition in tested assays, and Xrn1-mediated 5′-3′ decay limits accumulation of *Ty1* antisense RNAs that mediate trans-silencing [47].

Another key post-transcriptional modification of LTR-RTE RNA is m^6^A methylation, added by a complex of methyltransferases MTA and MTB along with specific cofactors in plants [48,49]. M^6^A regulates mRNA stability and translation in response to developmental and environmental signals [50]. In case of retrotransposons, m^6^A RNA methylation appears to be a double-edged sword across different organisms [51]. In human cells, m^6^A on *LINE-1* RNAs promotes translation and retrotransposition by recruiting eIF3 [52]. In contrast, plant studies suggest m^6^A marks on LTR-RTE transcripts generally lead to their sequestration in stress granules (SG) with reduced VLP assembly and cDNA production, rather than general decay, limiting translation [51]. This sequestration in SGs acts as a posttranscriptional repression mechanism by spatially isolating the *LTR-RTE* RNA and preventing it from proceeding through steps required for active retrotransposition such as VLP assembly and synthesis of cDNA [48]. Under heat stress in *Arabidopsis*, the RNA demethylase AtALKBH9B relocates to SG and removes m^6^A from *ONSEN* RNA [48]. The demethylation of *ONSEN* RNA is critical for releasing *ONSEN* RNA from SGs and enabling its mobilization (Figure 1). This dynamic methylation-demethylation balance serves as an epitranscriptomic control mechanism regulating *ONSEN* activity under stress conditions. Remarkably, some orchids have co-opted this strategy and a *Ty3*/*Gypsy* retrotransposon lineage encodes its own AlkB-type demethylase domain to erase m^6^A and boost its expression. This demethylase protein is encoded by additional ORFs and performs binding and demethylation of the LTR-RTE ssRNA on compatible levels with its *A. thaliana* homolog ALKBH9B [53]. This suggests that some LTR-RTEs may directly manipulate the host epigenetic machinery to promote their own transcription and mobilization.

## 3. Virus-like Particles Formation and Reverse Transcription

### 3.1. LTR-RTE Translation

Full-length LTR-RTEs typically encode five proteins: the Gag protein and the Pol polyprotein (cleaved by an aspartic protease/AP) which includes reverse transcriptase (RT), ribonuclease H (RH), and integrase (IN). Along with basal proteins, some families of LTR-RTEs can harbor additional Open Reading Frames (aORFs) which may confer regulatory functions. Recent studies suggest that aORFs can be found in most of the LTR-RTE lineages across diverse groups of eukaryotes, including plants [43,54,55], fungi [56,57,58] and animals [37,59,60,61].

The translation of LTR-RTE RNAs is entirely performed by the host cell translation machinery, and it depends on a number of ribosomal proteins. Indeed, among the host proteins promoting LTR-RTE transposition identified in yeast, many proteins were involved in translation including Bud21, Dbp7, Dfg10, Hcr1, Mrt4, Nat4, as well as numerous ribosomal proteins (e.g., Rpl7a, Rpl16b, Rpl19a, Rpl27a, Rpl31a, Rpl33b), ribosome biogenesis factors (e.g., Rsa3, Dpb7) and the translation initiation factor eIF2A [62]. These proteins support proper ribosome assembly and function, enabling robust translation of *Ty1* RNA. Most of these genes are conserved across eukaryotes, including plants, and some have been shown to promote retroviral or LTR-RTE replication, such as the RNA lariat debranching enzyme Dbr1 and NDR2 [62,63,64]. However, the mechanistic insights into this process remain unresolved. Moreover, the *Arabidopsis dbr1-1* null mutant exhibits embryonic lethality, which challenges experimental studies of DBR1’s role in the LTR-RTE life cycle [65]. However, viable hypomorphic alleles (*dbr1-2* and *dbr1-3*) have recently been described, exhibiting intron lariat accumulation and developmental phenotypes (curly leaves, increased branching) without lethality, opening new possibilities for elucidating DBR1’s role in LTR-RTE transcript processing and retrotransposition [65].

While GAG and POL proteins are often encoded by a single ORF, the LTR-RTE life cycle requires a significant excess of GAG protein over POL proteins. Two main strategies have been described for plants, animals, and yeast: translation from a single transcript via ribosomal frameshifting or encoding two transcripts, full-length genomic RNA and a short GAG isoform [18,66]. The latter mechanism is particularly well-characterized in the *A. thaliana EVADE* retrotransposon, where alternative splicing removes the entire protease domain and introduces a frameshift that creates a premature stop codon shortly after the splice junction, terminating GAG translation [27]. The resulting shGAG mRNA associates preferentially with polysomes for efficient translation, while the full-length GAG-POL transcript undergoes normal processing. Similar splicing-dependent shGAG production has been documented in barley *BARE1* elements, demonstrating conservation of this strategy across distant plant species [67].

### 3.2. Cytoplasmic Granules and VLP Formation

VLPs serve as reservoirs for reverse transcription and integrasome formation, representing key steps in the LTR-RTE life cycle. VLPs are assembled within specialized cytoplasmic foci known as retrosomes, which act as sites for VLP assembly [68]. Research on yeast retrotransposons *Ty1* and *Ty3* has identified host proteins involved in retrosome formation, VLP assembly, and maturation. These proteins play essential roles in the endoplasmic reticulum (ER) translocation machinery, the formation of distinct cytoplasmic granules, and RNA decapping and degradation processes. The VLP assembly begins when retrotransposon RNA is recognized by the signal recognition particle (SRP) during translation, resulting in the co-translational translocation of the nascent GAG protein into the ER. After the GAG protein is released into the cytoplasm, it associates with translating retrotransposon RNA on SRP–ribosome–nascent chain complexes, facilitating retrosome formation (Figure 1) [68]. Inhibition of ER-directed protein transfer using tunicamycin leads to increased accumulation of retrosomes, demonstrating a balance between directing retrotransposon proteins into the ER lumen and their retention in the cytosol [68].

In addition to retrosomes, other types of cytoplasmic granules associated with GAG proteins have been reported in yeast and plants [69,70]. These protein–RNA granules are generally categorized as SGs and processing bodies (P-bodies) [71,72]. Such interactions can sequester LTR-RTE RNAs away from translation and promote packaging. Key host proteins that mediate this process include P-body components such as Xrn1 (5′→3′ exonuclease), Lsm1 (Sm-like protein), Pat1 (translation repressor), and Dhh1 (DEAD-box helicase). These factors facilitate the shift of LTR-RTE RNA from active translation to a translationally repressed state, suitable for packaging into VLPs (Figure 1) [72]. In *A. thaliana*, the GAG protein of the *EVADE* forms cytoplasmic granules. These granules resemble those found in yeast and other systems and are linked to cellular stress responses, including the upregulation of SG-associated genes. This suggests that plant retrotransposon GAG proteins can form cytoplasmic aggregates analogous to retrosomes and stress granules, contributing to retrotransposon particle assembly and potentially influencing host stress response pathways [69]. Fan et al. provided mechanistic insight into the roles of SGs in the LTR-RTE life cycle [48]. They demonstrated that SGs serve as cytoplasmic hubs for *ONSEN* LTR-RTE RNA accumulation and processing under heat stress in *A. thaliana*, facilitating GAG and RNA interactions essential for VLP formation and retrotransposition. SGs may recruit LTR-RTE RNAs via plant YTH-domain readers like the ECT family, similarly to mammalian YTHDF proteins.

It is worth noting that cytoplasmic granules can also contribute to LTR-RTE silencing as LTR-RTE RNAs can associate with small interfering RNA (siRNA) bodies. This process is triggered by ribosome stalling due to unfavorable codon usage of LTR-RTE RNAs. This stalling promotes RNA truncation and directs these RNAs to siRNA bodies. The SGS3 protein, which interacts with RNA-dependent RNA polymerase 6 (RDR6), undergoes liquid–liquid phase separation mediated by its prion-like domains. Together, ribosome stalling and SGS3-driven phase separation initiate epigenetic silencing of transposons by enabling selective recognition and processing of transposon RNAs (Figure 1) [73].

Thus, cytoplasmic granules exhibit dual roles in LTR-RTE lifecycles in plants, balancing LTR-RTE mobility with LTR-RTE silencing.

### 3.3. Reverse Transcription Initiation

Following VLP assembly and the packaging of all essential proteins and LTR-RTE RNAs, the synthesis of complementary DNA (cDNA) of LTR-RTE is initiated. This stage critically depends on various host proteins and RNA molecules, several of which have been identified and functionally characterized.

The process of reverse transcription has been studied using the yeast *Ty1* element model [74,75]. Based on the conservation of structural elements, a similar model is presumably used for cDNA synthesis by plant LTR-RTEs [74]. The process starts with RNA priming. The priming step requires host partial tRNAs (tRFs) that bind to a specific region of LTR-RTE RNA called the primer binding site (PBS) located downstream of 5′LTR in most plant LTR-RTEs (Figure 1) [76,77]. Most often, tRFs originate from the initiator methionyl tRNA (tRNA_i_^Met^) that selectively packaged together with two copies of the retrotransposon RNA. The 3′ acceptor stem of the tRNA base-pairs with a complementary PBS immediately downstream of the 5′ LTR. Types of tRFs involved in priming differ between species [76,77,78]. It is worth noting that in plants, tRFs can also play a negative role in the LTR-RTE life cycle by triggering small RNA generation and silencing [79,80].

### 3.4. Strand Transfer Process

Similarly to retroviruses, the critical feature of LTR-RTEs reverse transcription is the process of strand transfer [75,81]. This process not only allows maintaining the identity of LTR sequences, but also facilitates the emergence of new variants of elements due to intermolecular recombination (if the VLP contains 2 different but related RNAs) [82]. The canonical model of reverse transcription of LTR-RTEs suggests two transfer events for both minus (−) and plus (+) strand strong stop DNAs (ssDNAs) [82]. The process of strand transfer during reverse transcription has also been indirectly demonstrated for plant LTR-RTEs retrotransposons. The first evidence that strand transfer processes occur for plant LTR-RTEs was discussed by studying complex sequences in *Triticeae* genomes [83]. Strand transfer was further demonstrated by recombination between endogenous *Tnt1* and mini-*Tnt1* transfer vectors [84]. Deletion analysis of the *Nicotiana tabacum Tto1* element demonstrated the real length of the LTR redundancy region (R region) required for strand transfer and provided the first model for this process for a plant LTR-RTE [85].

In addition to retrotransposon proteins, a recent study demonstrated the requirement for host alternative end-joining (alt-EJ) DNA repair pathway to mediate second strand transfer and subsequent synthesis. The alt-EJ factors (including *Polθ*, *XRCC1*, *Ligase 3* and *Fen1*) utilize microhomologies in the PBS regions to anneal the minus-strand cDNA and the newly synthesized plus strand (Figure 1) [7]. Unlike yeast, plants possess a bona fide alt-EJ pathway, often called microhomology-mediated end-joining (MMEJ) [86,87,88].

Interestingly, the first strand switch stage is also a hot point for recombination between RNA molecules in a single VLPs. As it was demonstrated for *ONSEN* elements, these recombination events frequently occurred between RNAs of different members of the same family and were even called ‘(pseudo)sexual reproduction’ [89]. Several categories of host proteins were shown to promote intermolecular recombination of plant RNA viruses [90], but there is no information available about host proteins that directly influence intermolecular recombination during reverse transcription of LTR-RTE. This knowledge gap could be addressed through targeted CRISPR screens of RNA helicase and chaperone families (DExD/H-box helicases (e.g., *AtRH20* in *A. thaliana*), HSP70/90) to identify host factors that enhance LTR-RTE recombination frequency during reverse transcription. This builds on established roles of these protein classes in plant RNA virus recombination [91].

### 3.5. Extrachromosomal Circular DNA Formations

It is a well-established fact that LTR-RTE linear cDNA is frequently circularized in plants and other organisms, generating extrachromosomal circular DNAs (eccDNAs) [18,21]. The eccDNAs have been used as a marker for studying the active mobilome in various plant species [18,21,92,93,94,95,96,97,98,99,100]. This fact confirms the conservation of the LTR-RTE cDNA circularization among distant eukaryotes, but in the case of plants, the exact molecular mechanisms of eccDNA formation are still to be explored. Studies on retroviruses and yeast showed that single-LTR (1-LTR) molecules can be formed via homologous recombination (HR) or reverse transcription errors. Two-LTR eccDNAs are produced via NHEJ [101,102]. The role of HR and NHEJ in eccDNA formation has been investigated in *S. cerevisiae* lines that were deficient in the corresponding genes (*RAD52*, *RAD50*, *RAD51*, *RAD54*, *RAD57*, *CDC9*). It was demonstrated that in these mutant lines, *Ty1* element transposition and the level of unincorporated DNA were significantly elevated [103]. Further studies demonstrated that single-LTR and autointegration-derived eccDNAs of *Ty1* in *S. cerevisiae* are formed independently of HR and DNA ligase IV (Dnl4)-mediated NHEJ. Whereas two-LTR eccDNAs require Dnl4 and become enriched only when integration is defective (Figure 1) [101]. A recent study of *D. melanogaster mdg4* and mouse *IAP* systems revealed that alt-EJ mediates a PBS-guided circularization that predominantly yields 1-LTR eccDNA and is essential for both eccDNA biogenesis and new insertions, with a transient non-covalently closed circular intermediate [7].

## 4. LTR-RTE Insertions

### 4.1. Insertion Site Selection

After nuclear import, the pre-integration complex (PIC) matures into the intasome, a nucleoprotein assembly composed of a tetramer of integrase molecules bound to the ends of double-stranded cDNA. The precise behaviors of the intasome and its interactions with nuclear proteins remain incompletely understood. Notably, LTR-RTEs show preferential insertion into specific genomic regions, which varies among taxonomic groups, ranging from superfamilies to individual elements. Initially, transposon distribution bias in plants was thought to arise from selection pressure purging insertions from genic areas due to their strong negative fitness effects. However, studies of induced transposition demonstrate that some LTR-RTEs actively integrate non-randomly into the host genome [104]. A well-studied example involves chromoviruses (*Ty3*/*Gypsy* elements) containing putative targeting domains (PTDs) divided into group I and II chromodomains and Centromeric Retrotransposon (CR) motifs, which differ structurally and functionally [105,106,107]. Group I chromodomains, especially prevalent in non-seed plant LTR-RTEs, feature a canonical aromatic cage resembling *Drosophila* HP1, enabling binding to methylated histone H3 lysine 9 (H3K9me), a hallmark of heterochromatin [108]. In contrast, group II chromodomains lack these aromatic residues and function by mechanisms yet to be elucidated. Similarly, CR motifs, despite lacking affinity for epigenetic marks, facilitate integration into centromeric regions, likely via unknown chromatin interactions [105]. Experimental fusion of CR and group II chromodomains to fluorescent proteins confirmed heterochromatin targeting [109]. Structural modeling predicted that the CR motif of a wheat centromeric retrotransposon integrase directly interacts with the centromeric histone variant CENH3, suggesting specific protein–protein interactions (Figure 1) [110].

Unlike *Ty3*/*Gypsy* elements, plant *Ty1*/*Copia* elements lack conserved domains directing integration. In yeast, *Ty1* integration upstream of RNA polymerase III genes is mediated by interaction between *Ty1* integrase C-terminus and the Pol III subunit AC40 (Figure 1) [111]. Selective integration of *Ty1*/*Copia* elements in *A. thaliana*, such as *EVADE* and *ONSEN*, shows a strong association with histone H2A.Z variants (Figure 1). Mutants lacking H2A.Z isoforms HTA9 and HTA11 display a significant reduction in *ATCOPIA93* insertions and a more random integration pattern across the genome. This supports a conserved role for H2A.Z in guiding integration sites, potentially to prevent disruption of essential genes, a phenomenon observed across plants and yeast [104]. The mechanisms may involve chromatin spatial conformation or yet-to-be-identified integrase docking interactions.

Thus, chromatin-associated proteins and their epigenetic modifications play critical roles in LTR-RTE integration by guiding the intasome to specific genomic regions. These chromatin interactions not only ensure integration specificity but may also help to preserve genome integrity by directing insertions away from essential genes.

### 4.2. Integrase-Independent Retrotransposition

The integrase-independent pathway was first comprehensively characterized in fission yeast using the *Tf1* retrotransposon. The authors discovered that even when IN activity was completely abolished, *Tf1* retained substantial insertion activity—approximately 5% of wildtype levels. This finding challenged the long-held notion that integrase was absolutely essential for LTR-RTE mobility. Transposition of the IN-defective *Tf1* most frequently occurred in regions partially homologous to *Tf1* sequences, including the PBS, PPT, and other full-length Tf elements [112]. This suggested that integrase-independent integration proceeds via homologous recombination mechanisms mediated by the Rad52 protein (Figure 1). A hallmark of integrase-independent retrotransposition is its propensity to create tandem insertions, observed across diverse organisms from yeast to *Drosophila* and plants [113]. Plant genomes, especially in pericentromeric regions, contain abundant tandem LTR-RTE arrays. While these tandems were typically attributed to unequal crossing over between LTRs, single-strand annealing (SSA)-mediated insertion at existing elements likely contributes to their formation. Notably, the Rad52 protein and the SSA DNA repair pathway are well-conserved in plants, providing the cellular machinery necessary for integrase-independent retrotransposition [114,115].

In summary, the integrase-independent pathway offers an alternative retrotransposition mechanism relying on Rad52-mediated homologous recombination and SSA, enabling insertion into homologous sequences and generating tandem repeats. This pathway expands our understanding of LTR-RTE mobility and suggests conserved mechanisms across eukaryotes.

## 5. The Future Perspectives

As discussed above, our current knowledge of host proteins that promote the LTR-RTE life cycle in plants remains very limited. Over the past few decades, most research has focused on elucidating various silencing mechanisms rather than promoting mechanisms acting on LTR-RTEs. Another issue is that the majority of studies have been conducted in only two model organisms—*Arabidopsis* and rice. While these species have been crucial for understanding LTR-RTE silencing mechanisms, their relatively small genomes provided limited insight into the diverse LTR-RTE transposition scenarios found in nature. Finally, the number of “model plant LTR-RTEs” that have been systematically investigated is quite small. To gain deeper insights into LTR-RTE biology, we propose developing transposition reporter systems to facilitate the efficient tracking of transposition events. Combining these reporter constructs with modern high-throughput genome editing technologies will enable the identification of host genes that either promote or restrict transposition. The resulting candidate genes can then be validated using advanced molecular approaches, including interactomics and mass spectrometry.

### 5.1. Reporter System for Studying LTR-RTEs

Transposition reporter systems enable real-time monitoring of transposition events at cellular or organismal levels. The application of reporters drove significant advances in elucidation of host factors promoting LTR-RTE transposition in yeast and fruit fly. The first LTR-RTE reporter was developed in the 1985 paper introducing the “retrotransposon” term [116]. The authors employed a simple reporter assay by inserting an intron into the *TyH3* element (Figure 2). Intron excision during the element’s life cycle and intronless *TyH3* copy generation provided definitive evidence for an RNA intermediate in transposition [116,117]. Subsequent studies utilized intronic sequences by introducing them into fluorescent protein genes incorporated within retrotransposon reporter constructs. The intron has an antisense orientation relative to the reporter protein sequence and a sense orientation relative to the retrotransposon sequence. The intron is spliced out during the retroelement life cycle, and the reporter gene becomes functional in the newly created retrocopy. This type of reporter has been utilized for a human *LINE-1* element (Figure 2). A gene encoding monomeric fluorescent timer (FT) disrupted by an intron demonstrated that *LINE-1* retrotransposition occurs predominantly during the S phase of the cell cycle [118]. In mice, GFP reporters embedded within *IAP* elements revealed that a deletion between GAG and POL regions increases retrotransposition efficiency five-fold when full-length proteins are co-expressed (Figure 2). Moreover, the deleted *IAP* element can induce expression of flanking host genes [119]. Another type of reporter called CLEVR has been recently developed [120]. In this reporter system, the Watermelon reporter (WM) gene is placed in the U3 region of the 3′LTR while the promoter of this gene is located in the U5 region of the 5′LTR. During cDNA synthesis, a template switch juxtaposes the promoter and reporter gene, activating expression. CLEVR reporter detects in vivo replication of the *gypsy* LTR-RTE in *Drosophila* (Figure 2). Using CLEVR, researchers have shown an age-dependent increase in the number of glial cells expressing *Ty3*/*Gypsy*-CLEVR [120]. To advance plant LTR-RTE research, the CLEVR principle could be adapted by designing analogous systems where reverse transcription-induced template switching of LTR-RTEs (e.g., *ONSEN* or *EVADE*) activates a fluorescent or selectable marker gene. This enables direct visualization and selection of transposition events in living plant cells. This would facilitate genome-wide screens for host factors influencing LTR-RTE transposition frequency.

In plants, the SAKE system for *A. thaliana* employs the *ONSEN* LTR-RTE with a luciferase gene inserted and disrupted by an artificial intron (Figure 2). Application of this reporter revealed that young, rapidly growing seedlings exhibit elevated luciferase signal, reflecting heightened *ONSEN* activity in proliferating tissues [121]. These examples demonstrate that reporter systems are valuable instruments for studying LTR-RTE biology in different species. As more active and inducible LTR-RTEs will be discovered in different plant species, more reporters can be designed and developed. The main limitation of reporter systems is their reliance on plant transgenesis—a labor-intensive and time-consuming process that remains challenging for the majority of non-model plants. Additionally, while intron-based reporters like SAKE successfully detect completed retrotransposition cycles, they lack the resolution to detect replication intermediates or to measure the efficiency of distinct steps of the LTR-RTE life cycle.

### 5.2. Novel Methods for Exploiting LTR-RTE Biology in Non-Model Plant Species

Numerous molecular techniques have been developed and successfully applied to study LTR-RTEs in model plant species. In recent years, methodological advances have yielded novel approaches that are more easily applicable to non-model plant species. This significantly expands the potential for investigating LTR-RTE biology in plants with large and complex genomes. Notably, new interatomic and genome-editing approaches offer a powerful toolbox to elucidate the molecular mechanisms underlying LTR-RTE life cycle.

A range of modern methods for elucidating protein–protein interactions (PPIs) has emerged, providing new strategies for understanding complex PPI networks associated with LTR-RTEs in plant cells. Among these, the proximity labeling system (e.g., TurboID) represents a particularly promising approach (Figure 3). TurboID enables the identification of protein partners through proximity-dependent biotin labeling, achieved by fusing a target protein with a mutated version of *Escherichia coli* biotin ligase (TurboID or miniTurbo) [122]. Biotinylated proteins in proximity to the fusion partner can then be purified and identified using mass spectrometry. This method has been successfully implemented in plant systems, proving to be fast, sensitive, and broadly applicable in planta [122].

In the context of LTR-RTEs, TurboID could be applied by creating fusions with integrase or GAG proteins, which are known to interact with various host cellular components during the transposition cycle. Such an approach would allow mapping of the dynamic protein networks associated with LTR-RTE assembly, movement, and regulation. Furthermore, TurboID has been effectively used to profile proteomes of specific cellular compartments by directing the biotin ligase to compartment-specific proteins. This strategy could also be adapted to detect LTR-RTE and host protein interactions within subcellular structures such as stress granules or P-bodies. These structures are believed to form transiently during the LTR-RTE life cycle and may play regulatory roles in transposition control. In parallel, advances in computational biology provide additional avenues for exploring LTR-RTE-related interactions. Structure-based prediction tools such as AlphaFold, along with other bioinformatic approaches (e.g., co-expression analysis of host genes with active LTR-RTEs, GWAS, phylogenetic profiling, etc.), can be employed to model and predict potential LTR-RTE–host protein interfaces [123]. Although such computational predictions require experimental confirmation, they serve as valuable starting points for hypothesis generation and targeted validation. For instance, this combined approach has been used to investigate the interaction between centromeric retrotransposon (CR) motifs and the centromeric histone CENH3 protein [110].

Together, these emerging methodological advances—integrating proximity labeling, proteomics, and computational modeling—offer an increasingly comprehensive framework for understanding the molecular interplay between LTR-RTEs and their host cellular environments.

Active transposon silencing in plant genomes poses a major challenge for investigating the functional aspects of the LTR-RTE life cycle. Without transposition activity, deciphering the biological roles of TE-encoded proteins remains limited. Artificial activation is critical for elucidating the molecular mechanisms underlying transposon biology and for characterizing both host and LTR-RTE proteins that participate in the life cycle. Several strategies have been developed to artificially activate LTR-RTEs and overcome epigenetic silencing constraints [124]. A recent study introduced a promising approach for targeted transcriptional activation of LTR-RTEs [125]. Using a CRISPR-based system, the expression of the *EVADE* was precisely activated via a catalytically inactive Cas9 (dCas9) fused to the SunTag-VP64 transcriptional activator complex (Figure 3). This method resulted in a dramatic upregulation of *EVADE* transcription by several hundred- to thousand-fold, demonstrating the effectiveness of programmable activation of silent LTR-RTEs. However, whether this transcriptional activation leads to actual *EVADE* transposition events remains to be experimentally verified. Combining this dCas9-SunTag activation approach with a transposition reporter represents a critical experimental strategy for future investigation of the complete LTR-RTE life cycle, enabling precise, programmable activation of LTR-RTEs to determine the functional interactome and elucidate retrotransposition mechanisms.

Generating knockout mutants in key transposon silencing genes continues to be the most powerful strategy for studying LTR-RTE reactivation and mobility, although this approach can be time-consuming and expensive for most non-model plant species. The recent advent of modern genome editing techniques has significantly simplified mutant generation. Innovative in planta delivery systems, such as virus-induced genome editing (VIGE) [126,127,128,129] and Cut–Dip–Budding methodologies [130,131], enable the direct delivery of genome editing components into meristematic tissues (Figure 3). These systems facilitate efficient and heritable genome modifications while bypassing traditional tissue culture and transformation steps. These approaches allow for faster generation of mutant plants while reducing the negative effects of tissue culture on genome stability and spontaneous LTR-RTE activation. Applying such approaches to generate mutations in key genes involved in transposon silencing and LTR-RTE activation allows in-depth functional analyses of LTR-RTEs across a broad range of plant species (Figure 3). Expanding experimental studies beyond traditional model systems is likely to reveal new aspects of LTR-RTE regulation and activity that are specific to complex genomes.

Genome editing can also be used to experimentally verify candidate host factors associated with LTR-RTE copy number variation identified in natural *Arabidopsis* ecotypes [132,133]. These candidate loci represent a valuable resource for dissecting the genetic basis of LTR-RTE biogenesis and control in plants. Establishing direct causal relationships between these genes and LTR-RTE activity will require reverse genetics approaches, including multiplex genome editing and the development of plant populations carrying targeted mutations. Importantly, high-throughput CRISPR-based screening methods that enable simultaneous mutagenesis of thousands of genes in multiple combinations hold considerable promise for systematically identifying the host determinants of TE mobility [134].

## 6. Concluding Remarks

Our understanding of the LTR-RTE life cycle and the host proteins involved in this process remains disproportionately limited, given the key role of LTR-RTEs in plant genome evolution and breeding. Recent studies across different organisms have revealed that the LTR-RTE life cycle is deeply integrated into host cellular networks. Technological advances in protein interaction assays, genome editing, and transcriptional activation are poised to enable functional investigations of LTR-RTEs and their associated host proteins in non-model plants. Integrating these approaches with LTR-RTE reporters, high-throughput screening, and proteomic techniques will not only deepen our understanding of LTR-RTE biology but also shed light on broader principles of genome plasticity, adaptation, and evolution in plants.

## Figures and Tables

**Figure 1 ijms-27-00374-f001:**
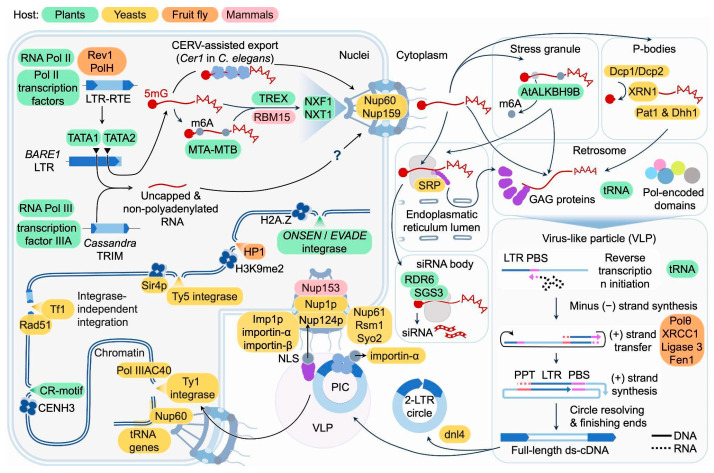
Schematic overview of cellular components that support key stages of the LTR-RTE life cycle in plants (green), in yeasts (*Saccharomyces cerevisiae* and *Schizosaccharomyces pombe*) (yellow), fruit fly (orange), and mammals (pink). In the nucleus, RNA polymerase complexes, transcription factors, chromatin-associated proteins, RNA-modifying enzymes, and export machineries promote LTR-RTE transcription, RNA processing, and nuclear export, while integrase partners and chromatin features influence integration site selection. In the cytoplasm, RNA granule components, RNA-binding proteins, small-RNA factors, and translation/targeting machinery regulate retrotransposon RNA stability, translation, and virus-like particle (VLP) assembly. Within VLPs and during nuclear re-entry, host DNA repair, polymerase, ligase, import, and nuclear pore complexes cooperate with Pol-encoded activities to complete reverse transcription, pre-integration complex trafficking, and chromosomal integration of newly synthesized cDNA.

**Figure 2 ijms-27-00374-f002:**
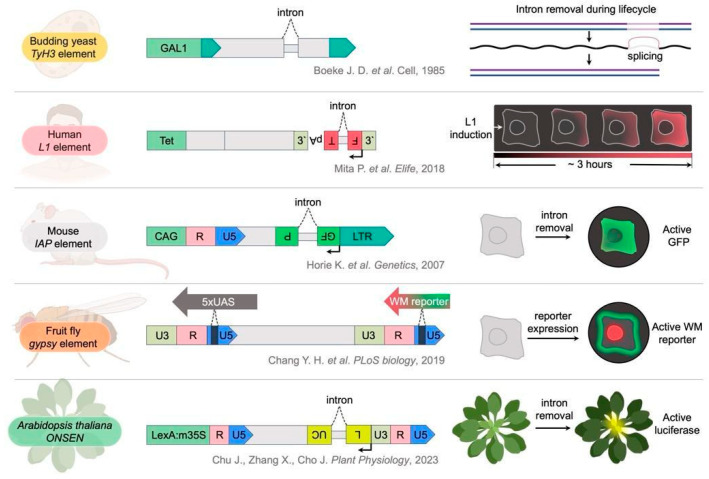
Reporter systems for the detection of retrotransposon transposition events in different organisms. The right panel illustrates reporter gene states before and after retrotransposition events. The yeast Ty3-based reporter system exploits intron splicing during the retrotransposon’s RNA intermediate phase [116]. Comparable reporter constructs derived from human *LINE-1*, murine intracisternal A-particle (*IAP*), and *Arabidopsis thaliana ONSEN* (SAKE) retroelements also utilize intron splicing to rescue reporter expression, thus enabling visualization and quantification of transposition [118,119,120,121]. The CLEVR system of the fruit fly uniquely employs the capacity of retrotransposons to undergo template switching during cDNA synthesis, resulting in promoter repositioning that activates the reporter gene.

**Figure 3 ijms-27-00374-f003:**
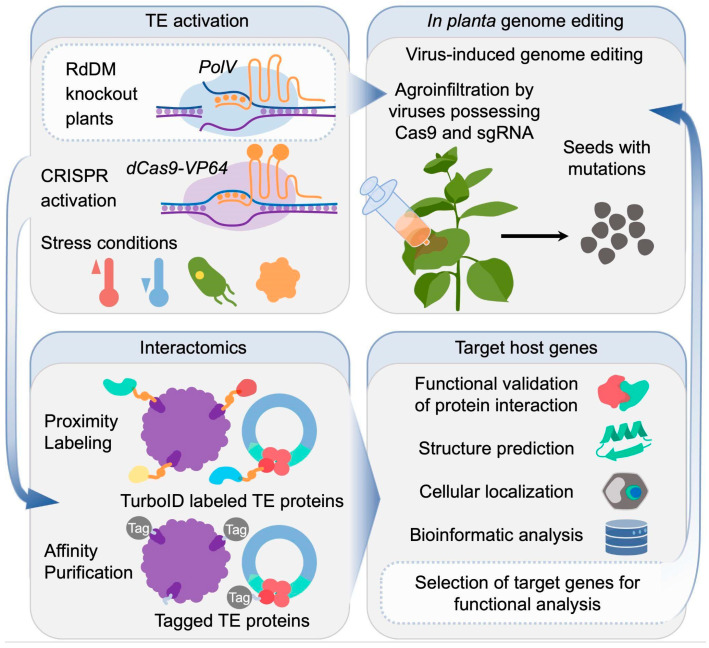
Integrated approach to identify proteins involved in the LTR-RTE life cycle in plants using transposable element activation tools, interactomics methods, bioinformatic characterization of target genes, and in planta virus-induced genome editing.

## Data Availability

No new data were created or analyzed in this study. Data sharing is not applicable to this article.

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
