# Peer review of "Host Factors Promoting the LTR Retrotransposon Life Cycle in Plant Cells: Current Knowledge and Future Directions"

_ijms, 2025, doi:10.3390/ijms27010374_

Round 1

Reviewer 1 Report

Comments and Suggestions for Authors

Comments to the author

The manuscript entitled "Ubiquitous Yet Underexplored: Missing Host Factors Promoting the LTR Retrotransposon Life Cycle in Plant Cells" focuses on host factors that promote the life cycle of LTR retrotransposons (LTR-RTEs), an underexplored yet crucial aspect of transposon biology. The manuscript is clearly written, with a rigorous logical structure organized according to the LTR-RTE life cycle, and supported by extensive and up-to-date literature. However, I believe the review requires some revisions prior to publication. First, it reads more like a general overview of the LTR-RTE life cycle than a focused synthesis on host promoting factors. The authors should emphasize these factors and build the discussion structure specifically around them. Furthermore, I suggest incorporating more examples and discussions from plant systems, along with updates to relevant papers. Currently, most supporting evidence is derived from yeast, Drosophila, and mice.

Author Response

We thank the reviewer for the comments. Please, find our point-by-point answers below. We believe these revisions have elevated the manuscript’s clarity and impact. 

Q1: First, it reads more like a general overview of the LTR-RTE life cycle than a focused synthesis on host promoting factors. The authors should emphasize these factors and build the discussion structure specifically around them. 

A1: We have substantially revised the manuscript to enhance clarity and focus specifically on host factors that promote the LTR retrotransposon life cycle in plant systems. The discussion has been reorganized to center around these host-promoting factors. Additionally, Figure 1 has been redesigned to emphasize plant-specific elements. We also reduced the inclusion of examples from non-plant systems to maintain a clear and relevant scope. We also changed the title of the paper to make it more focused on future directions.

Q2: Furthermore, I suggest incorporating more examples and discussions from plant systems, along with updates to relevant papers. Currently, most supporting evidence is derived from yeast, Drosophila, and mice.

A2: Yes, we agree that our review includes more examples of host proteins from non-plant organisms. Unfortunately, data on proteins that facilitate the complete LTR-RTE life cycle in plants remain very limited. Although there are many reviews on transposable element silencing systems in plants, silencing is not the focus of our current review. We would welcome including more plant-specific papers, but to our knowledge, we have already covered most of the available studies in the plant field.

Reviewer 2 Report

Comments and Suggestions for Authors

Introduction, the statement "Novel insertions arise in the final stages of the retrotransposon life cycle" is slightly misleading. While integration is the final step for a single cycle, the life cycle is a continuum. Rephrasing this to clarify that novel insertions are the outcomeof the completed cycle would improve precision.

The caption for Figure 1 is incomplete. It states "The color of each rectangle indicates the host organism: green for plants," but cuts off. The full color key must be provided to make the figure interpretable. The caption should be a self-contained legend explaining all symbols and colors used.

In section 2.1, the description of the Fun30 remodelers in yeast is detailed, but the direct connection to plant systems is not explicitly made. A sentence should be added to speculate on the existence or nature of potential functional homologs of Fun30 in plants and their possible role in LTR-RTE regulation, to better bridge the knowledge gap the review aims to address.

The transition between sections 2.1 and 2.2 is abrupt. The paragraph on the "retrotranscriptome" is interesting but feels isolated. It would be more impactful if integrated into the discussion on transcription or used to introduce the concept of RNA fate, which leads naturally into the topics of export and processing.

In section 2.2, the statement "Direct studies on the nuclear export of plant LTR-RTE RNAs are lacking" is accurate but is followed by extensive detail from other systems. To enhance the critical analysis, the authors should explicitly compare the known export mechanisms (e.g., NXF1/TREX, RTE/MTE) and propose which pathways are most likely to be operational in plants based on genomic features, making this section more hypothesis-driven.

The discussion on Dbr1 in section 2.3 notes the embryonic lethality of the Arabidopsis dbr1mutant as a challenge. This is a crucial point. The authors should suggest potential alternative approaches to study Dbr1's role in plants, such as conditional knockdown systems or the use of hypomorphic alleles, which would strengthen the "Future Perspectives" theme.

Section 3.2 provides a good description of cytoplasmic granules but could be more critical. The dual role of granules (promoting packaging vs. initiating silencing) is a key concept. A specific paragraph contrasting the pro-retrotransposition role of P-bodies in yeast with the anti-retrotransposition role of SGs/siRNA bodies in plants would provide a clearer and more nuanced synthesis for the reader.

The mention of SGS3 and phase separation in the context of siRNA bodies is a modern and relevant point. However, its connection to the promotion of the LTR-RTE life cycle is negative. The authors should clearly frame this within the review's focus, perhaps by stating that understanding these antagonistic host factors is equally important for a complete picture of the host-retrotransposon interface.

In section 3.4, the sentence "This represents a significant gap in our understanding" is a general statement. It would be more valuable to propose a specific, testable hypothesis regarding which host proteins (e.g., RNA helicases, chaperones) might influence intermolecular recombination during reverse transcription in plants.

Figure 2 is a valuable illustration of reporter systems. However, the caption could be more informative. For each system (TyH3, LINE-1-FT, IAP, CLEVR, SAKE), a brief one-sentence explanation of the principleof detection (e.g., intron splicing, promoter repositioning) would make the figure more accessible to non-specialists.

In section 5.1, the SAKE system is described. The manuscript should mention a key limitation of this and similar intron-based reporters: they detect completed retrotransposition cycles but may not easily allow for the isolation of intermediates or the assessment of the efficiency of individual steps (e.g., VLP formation, reverse transcription).

The description of the CLEVR system is excellent. However, the authors could propose a direct adaptation of this clever "template-switch-activates-reporter" principle for designing similar systems for specific plant LTR-RTEs, which would be a constructive suggestion.

The discussion on CRISPR activation (dCas9-SunTag) in section 5.2 is a strong point. The authors correctly note that transposition was not confirmed. This should be framed as a critical future experiment, suggesting that such a system, combined with a reporter, could definitively link transcriptional activation to successful transposition.

While VIGE and Cut-Dip-Budding are appropriately mentioned, the text could briefly elaborate on their specific advantage for studying LTR-RTEs: they potentially allow for the generation of mutant plants without the confounding effects of tissue culture, which is known to induce transposon activity, thus providing a cleaner genetic background.

The manuscript would benefit from a dedicated short section or paragraph on computational/bioinformatic approaches. Beyond AlphaFold, methods for predicting LTR-RTE-host protein interactions (e.g., co-expression analysis of host genes with active LTR-RTEs, phylogenetic profiling) could be suggested as a means to generate candidate host factors for validation.

Throughout the manuscript, some terminology could be more precisely defined for a broad audience. For example, "alt-EJ" is used before it is fully explained. While it is defined later, ensuring that acronyms are defined at first use in each major section would improve readability.

The conclusion of the manuscript (implicit in the "Future Perspectives" section) is effective but could be more forceful. A standalone "Concluding Remarks" paragraph should be added to succinctly summarize the main message: that LTR-RTEs are master integrators of host cellular networks, and that leveraging new tools in non-model plants will be key to uncovering the unique aspects of plant-retrotransposon interactions.

The reference list is comprehensive but should be checked for consistency with the journal's formatting guidelines. Some references use "et al." after 10 authors, while others list more; this should be standardized according to IJMS rules (typically "et al." for more than 10 authors).

A minor point: on page 1, the author list includes "Alexander Soloviev" while the affiliation list has "Alexander A. Soloviev". The names must be consistent between the author list and the affiliations to avoid confusion.

Author Response

We thank the reviewer for the thorough and constructive evaluation of this manuscript. We sincerely appreciate your excellent work as a reviewer. Your detailed comments have substantially improved the clarity, precision, and scope of this review. Your feedback has ensured that this review provides a more comprehensive and nuanced view of LTR-RTE biology in plants. 

Q1: Introduction, the statement "Novel insertions arise in the final stages of the retrotransposon life cycle" is slightly misleading. While integration is the final step for a single cycle, the life cycle is a continuum. Rephrasing this to clarify that novel insertions are the outcome of the completed cycle would improve precision.

A1: Corrected.

Q2: The caption for Figure 1 is incomplete. It states "The color of each rectangle indicates the host organism: green for plants," but cuts off. The full color key must be provided to make the figure interpretable. The caption should be a self-contained legend explaining all symbols and colors used.

A2: Figure 1 has been modified, and the figure legend has been expanded to provide a comprehensive, self-contained description.

Figure 1. Host factors promoting the LTR retrotransposon life cycle in plants, yeast, fruit fly and mammals..

Schematic overview of cellular components that support key stages of the (LTR‑RTE) life cycle in plants (green) and in yeasts (Saccharomyces cerevisiae and Schizosaccharomyces pombe) (yellow), fruit fly (orange), and mammals (pink). In the nucleus, RNA polymerase complexes, transcription factors, chromatin-associated proteins, RNA-modifying enzymes, and export machineries promote LTR‑RTE transcription, RNA processing, and nuclear export, while integrase partners and chromatin features influence integration site selection. In the cytoplasm, RNA granule components, RNA-binding proteins, small-RNA factors, and translation/targeting machineries regulate retrotransposon RNA stability, translation, and virus-like particle (VLP) assembly. Within VLPs and during nuclear re-entry, host DNA repair, polymerase, ligase, import, and nuclear pore complexes cooperate with Pol-encoded activities to complete reverse transcription, pre-integration complex trafficking, and chromosomal integration of newly synthesized cDNA.

Q3: In section 2.1, the description of the Fun30 remodelers in yeast is detailed, but the direct connection to plant systems is not explicitly made. A sentence should be added to speculate on the existence or nature of potential functional homologs of Fun30 in plants and their possible role in LTR-RTE regulation, to better bridge the knowledge gap the review aims to address.

A3: We agree. We added the following paragraph to the MS: 

Plants also possess Snf2-like chromatin remodelers, such as DDM1, which maintains DNA and histone methylation in heterochromatic regions by regulating nucleosome occupancy and composition, thereby influencing LTR-RTE transcription [https://pmc.ncbi.nlm.nih.gov/articles/PMC10529913/]. Similarly, the INO80 chromatin remodeling complex, recently characterized in rice (OsINO80), promotes H3K27me3 and H3K9me2 deposition to maintain LTR-RTE silencing [https://pmc.ncbi.nlm.nih.gov/articles/PMC11686384/]. Unlike yeast Fun30 remodelers (Fft2/Fft3), which position nucleosomes at the U3 LTR region to control TSS selection and produce truncated non-functional transcripts, plant mechanisms primarily involve nucleosome composition changes and overall occupancy levels that support DNA methylation and histone modifications rather than TSS control. Thus, chromatin remodeling complexes regulate LTR-RTEs across eukaryotes, but in plants they primarily function in silencing rather than promoting transposition.

Q4: The transition between sections 2.1 and 2.2 is abrupt. The paragraph on the "retrotranscriptome" is interesting but feels isolated. It would be more impactful if integrated into the discussion on transcription or used to introduce the concept of RNA fate, which leads naturally into the topics of export and processing.

A4: We modified this part and decided to exclude “retrotranscriptome” part from the MS.

Q5: In section 2.2, the statement "Direct studies on the nuclear export of plant LTR-RTE RNAs are lacking" is accurate but is followed by extensive detail from other systems. To enhance the critical analysis, the authors should explicitly compare the known export mechanisms (e.g., NXF1/TREX, RTE/MTE) and propose which pathways are most likely to be operational in plants based on genomic features, making this section more hypothesis-driven.

A5: We agree and significantly modified this part.We have added a brief comparison of mRNA export features in plants, animals, and yeast, highlighting the possibility of plant retrotransposon mRNA export via pathways similar to those in animals.

Q6: The discussion on Dbr1 in section 2.3 notes the embryonic lethality of the Arabidopsis dbr1 mutant as a challenge. This is a crucial point. The authors should suggest potential alternative approaches to study Dbr1's role in plants, such as conditional knockdown systems or the use of hypomorphic alleles, which would strengthen the "Future Perspectives" theme.

A6: We have added this part to the MS: 

However, viable hypomorphic alleles (dbr1-2 and dbr1-3) have recently been described, exhibiting intron lariat accumulation and developmental phenotypes (curly leaves, increased branching) without lethality, opening new possibilities for elucidating DBR1’s role in LTR-RTE transcript processing and retrotransposition [https://doi.org/10.1101/2023.07.25.550414].

Q7: Section 3.2 provides a good description of cytoplasmic granules but could be more critical. The dual role of granules (promoting packaging vs. initiating silencing) is a key concept. A specific paragraph contrasting the pro-retrotransposition role of P-bodies in yeast with the anti-retrotransposition role of SGs/siRNA bodies in plants would provide a clearer and more nuanced synthesis for the reader.

A7: Thank you for this excellent suggestion. We have added the following paragraph at the end of this section of the MS :

‘Thus, cytoplasmic granules exhibit dual roles in LTR-RTE life cycles in plants balancing LTR-RTE mobility with LTR-RTE silencing’.

Q8: The mention of SGS3 and phase separation in the context of siRNA bodies is a modern and relevant point. However, its connection to the promotion of the LTR-RTE life cycle is negative. The authors should clearly frame this within the review's focus, perhaps by stating that understanding these antagonistic host factors is equally important for a complete picture of the host-retrotransposon interface.

A8: We have added this sentece to the MS: “It is worth noting that cytoplasmic granules can also contribute to LTR-RTE silencing as LTR-RTE RNAs can associate with small interfering RNA (siRNA) bodies. “

Q9: In section 3.4, the sentence "This represents a significant gap in our understanding" is a general statement. It would be more valuable to propose a specific, testable hypothesis regarding which host proteins (e.g., RNA helicases, chaperones) might influence intermolecular recombination during reverse transcription in plants.

A9: The following paragraph has been added to the MS:

This knowledge gap could be addressed through targeted CRISPR screens of RNA helicase and chaperone families (DExD/H-box helicases (e.g. AtRH20 in A. thaliana), HSP70/90) to identify host factors that enhance LTR-RTE recombination frequency during reverse transcription, building on established roles of these protein classes in plant RNA virus recombination [https://pubmed.ncbi.nlm.nih.gov/25693185/].

Q10: Figure 2 is a valuable illustration of reporter systems. However, the caption could be more informative. For each system (TyH3, LINE-1-FT, IAP, CLEVR, SAKE), a brief one-sentence explanation of the principle of detection (e.g., intron splicing, promoter repositioning) would make the figure more accessible to non-specialists.

A10: We thank the reviewer for this comment. We have added more information to the Figure’s capture.

Q11: In section 5.1, the SAKE system is described. The manuscript should mention a key limitation of this and similar intron-based reporters: they detect completed retrotransposition cycles but may not easily allow for the isolation of intermediates or the assessment of the efficiency of individual steps (e.g., VLP formation, reverse transcription).

A11: Thank you for this suggestion. The information has been added to the MS: ‘Additionally, while intron-based reporters like SAKE successfully detect completed retrotransposition cycles, they lack the resolution to detect replication intermediates or measure the efficiency of distinct steps of LTR-RTE life cycle.

Q12: The description of the CLEVR system is excellent. However, the authors could propose a direct adaptation of this clever "template-switch-activates-reporter" principle for designing similar systems for specific plant LTR-RTEs, which would be a constructive suggestion.

A12: We agree. The following text has been added to the MS: 

To advance plant LTR-RTE research, the CLEVR principle could be adapted by designing analogous systems where reverse transcription-induced template switching of LTR-RTEs (e.g. ONSEN or EVD) activates a fluorescent or selectable marker gene, enabling direct visualization and selection of recombination events in living plant cells and facilitating genome-wide screens for host factors influencing LTR-RTE recombination frequency.

Q13: The discussion on CRISPR activation (dCas9-SunTag) in section 5.2 is a strong point. The authors correctly note that transposition was not confirmed. This should be framed as a critical future experiment, suggesting that such a system, combined with a reporter, could definitively link transcriptional activation to successful transposition.

A13: We agree, thank you. The following text has been added: “Combining this dCas9-SunTag activation approach with a transposition reporter represents a critical experimental strategy for future investigation of the complete LTR-RTE life cycle, enabling precise, programmable activation of LTR-RTEs to determine the functional interactome and elucidate retransposition mechanisms.

Q14: While VIGE and Cut-Dip-Budding are appropriately mentioned, the text could briefly elaborate on their specific advantage for studying LTR-RTEs: they potentially allow for the generation of mutant plants without the confounding effects of tissue culture, which is known to induce transposon activity, thus providing a cleaner genetic background.

A14: The following text has been added: “These approaches allow for faster generation of mutant plants while reducing the negative effects of tissue culture on genome stability and spontaneous TE activation

Q15: The manuscript would benefit from a dedicated short section or paragraph on computational/bioinformatic approaches. Beyond AlphaFold, methods for predicting LTR-RTE-host protein interactions (e.g., co-expression analysis of host genes with active LTR-RTEs, phylogenetic profiling) could be suggested as a means to generate candidate host factors for validation.

A15: We agree that these computational and bioinformatic approaches are very important, but this topic is somewhat beyond the scope of the present review, which is primarily focused on experimental strategies for dissecting LTR-RTE functions and host interactions. Nevertheless, we now briefly mention co-expression analysis, phylogenetic profiling, and structure-based interaction prediction as complementary avenues to generate candidate host factors for future experimental validation.

Q16: Throughout the manuscript, some terminology could be more precisely defined for a broad audience. For example, "alt-EJ" is used before it is fully explained. While it is defined later, ensuring that acronyms are defined at first use in each major section would improve readability.

A16: We check the MS and made correction throughout the text.

Q17: The conclusion of the manuscript (implicit in the "Future Perspectives" section) is effective but could be more forceful. A standalone "Concluding Remarks" paragraph should be added to succinctly summarize the main message: that LTR-RTEs are master integrators of host cellular networks, and that leveraging new tools in non-model plants will be key to uncovering the unique aspects of plant-retrotransposon interactions.

A17: The section has been added.

Q18: The reference list is comprehensive but should be checked for consistency with the journal's formatting guidelines. Some references use "et al." after 10 authors, while others list more; this should be standardized according to IJMS rules (typically "et al." for more than 10 authors).

A18: We checked the reference and formatted according to the IJMS rules.

Q19: A minor point: on page 1, the author list includes "Alexander Soloviev" while the affiliation list has "Alexander A. Soloviev". The names must be consistent between the author list and the affiliations to avoid confusion.

A19: Thank you! This is technical error.

Reviewer 3 Report

Comments and Suggestions for Authors

 The manuscript “Ubiquitous Yet Underexplo red: Missing Host Factors Promoting the LTR Retrotransposon Life Cycle in Plant Cells” describes analysis of Long Terminal Repeat (LTR) retrotransposons (LTR-RTEs) as dominant component of plant genomes, comprising up to 90% of genomic DNA in some species. Their extensive amplification over evolutionary time has been a major driver of genome diversification, influencing both plant evolution and breeding potential. These elements rely on the host transcriptional machinery for RNA production, which then undergoes processing. This review addresses the central question: What is known about host proteins that promote LTR retrotransposon transposition? Here, we describe the currently identified host factors and cellular compartments involved in each step. However, little is known about the precise mechanisms of action of these factors, and information on plants remains particularly limited, despite their exceptionally rich repertoire of retrotransposons. Authors proposed combining plant mobilomics with transposition reporter systems and advanced tools from genome editing, synthetic biology, and interactomics to elucidate plant-specific mechanisms.

Despite the good quality of writing, some text fragments need improvement to avoid any possible misunderstanding:

For exampe:

1) Direct studies on the nuclear export of plant LTR-RTE RNAs are lacking.

Suggested: There is a lack of direct studies on the nuclear export of long terminal repeat-retrotransposon RNAs (LTR-RTE RNAs) in plants.

2) Since most retrotransposons do not contain introns (except, for example, the Ogre elements in M. truncatula), multiple pathways are possible for recruiting export factors to its mRNA

Suggested: Since most retrotransposons do not contain introns, except for some examples such as the Ogre elements in M. truncatula, there are multiple pathways for recruiting export factors to their mRNA.

3) For instance, m⁶A-modified LTR-RTE RNAs – such as ONSEN transcripts demethylated by AtALKBH9B – may influence SG recruitment through plant YTH-domain reader proteins (ECT family), paralleling the functionality of mammalian YTHDF proteins

Suggested: For example, m6A-modified long non-coding RNAs (LTR-RNEs) - such as ONSEN transcripts demethylated by AtALKBH9B - may influence the recruitment of SGs through plant YTH domain reader proteins (ECT family), similar to the functionality of mammalian YTHDF proteins.

4) Studies on retroviruses and  yeast showed that one LTR (1-LTR) molecules can be formed via homological recombination or reverse transcription errors, 2-LTR circles produced due to NHEJ, and both variants can be result of autointegration.

Suggested: Studies on retroviruses and yeast have shown that single-copy LTR (is it  correct?) (1-LRT) molecules can be formed through homologous recombination or reverse transcription errors. Double-copy LTRs (2-LRTs) are produced through NHEJ (non-homologous end joining), and both variants can result from unintentional integration.

Comments on the Quality of English Language

Despite the good quality of writing, some text fragments need improvement to avoid any possible misunderstanding. See above. 

Author Response

We sincerely thank the reviewer for the valuable and constructive comments that helped improve the quality and clarity of our manuscript. Please, find our point-by-point answers below. We believe these revisions have elevated the manuscript’s clarity and impact. 

Q1: ‘Direct studies on the nuclear export of plant LTR-RTE RNAs are lacking’.

A1: Corrected.

Q2: ‘Since most retrotransposons do not contain introns (except, for example, the Ogre elements in M. truncatula), multiple pathways are possible for recruiting export factors to its mRNA’

A2: Corrected.

Q3: For instance, m⁶A-modified LTR-RTE RNAs – such as ONSEN transcripts demethylated by AtALKBH9B – may influence SG recruitment through plant YTH-domain reader proteins (ECT family), paralleling the functionality of mammalian YTHDF proteins Suggested: For example, m6A-modified long non-coding RNAs (LTR-RNEs) - such as ONSEN transcripts demethylated by AtALKBH9B - may influence the recruitment of SGs through plant YTH domain reader proteins (ECT family), similar to the functionality of mammalian YTHDF proteins.

A3: Corrected.

Q4: Studies on retroviruses and  yeast showed that one LTR (1-LTR) molecules can be formed via homological recombination or reverse transcription errors, 2-LTR circles produced due to NHEJ, and both variants can be result of autointegration.

Suggested: Studies on retroviruses and yeast have shown that single-copy LTR (is it  correct?) (1-LRT) molecules can be formed through homologous recombination or reverse transcription errors. Double-copy LTRs (2-LRTs) are produced through NHEJ (non-homologous end joining), and both variants can result from unintentional integration.

A4: Corrected.

Reviewer 4 Report

Comments and Suggestions for Authors

Dear Authors,

Thank you for submitting your review manuscript on “Ubiquitous Yet Underexplored: Missing Host Factors Promoting the LTR Retrotransposon Life Cycle in Plant Cells.” The topic is timely and valuable; however, several key areas require improvement to strengthen the manuscript’s clarity, coherence, and scientific depth.

Major Points to Improve

1. Sharpen the central narrative:

The manuscript is rich in information, but the main storyline is sometimes diluted by excessive methodological detail. Please clarify the review’s central question and ensure each section explicitly contributes to answering it.

2. Improve structure and flow:

Some subsections contain highly technical paragraphs that could be streamlined or reorganized. Adding clear transitional sentences between stages of the LTR-RTE life cycle will help readers follow the logic.

3. Balance cross-kingdom comparisons:

While comparisons with yeast, Drosophila, and mammals are informative, the plant-specific insights should be more emphasized. Several sections rely heavily on non-plant systems; consider guiding the reader more explicitly on how these findings translate to plants.

4. Condense overly detailed descriptions:

Certain mechanistic descriptions (e.g., nuclear export, decapping, strand transfer) are excessively long for a review. Summarize where possible and highlight only what is necessary to build your argument.

5. Clarify hypotheses vs. demonstrated evidence:

Occasionally, speculative statements appear alongside confirmed data. Please distinguish hypotheses, assumptions, or unverified mechanisms clearly.

6. Add a stronger critical analysis:

The review is comprehensive but largely descriptive. Consider incorporating critical evaluation of existing gaps, contradictions, and limitations in current plant LTR-RTE research.

7. Enhance figures for readability:

Figure 1 is informative but visually dense. Simplifying color coding or increasing text clarity could improve comprehension.

8. Future perspectives need more precision:

The proposed research directions are excellent but could benefit from clearer prioritization—what approaches are immediately feasible for plants vs. long-term innovations?

Minor Points

Ensure consistency of terminology, especially regarding “host factors,” “LTR-RTE,” “retrotranscriptome,” and plant-specific nomenclature.

Some sections would benefit from shorter sentences to aid readability.

A more concise abstract would increase its impact.

Improve the integration of citations by ensuring each statement is appropriately supported.

Comments on the Quality of English Language

The English language still needs additional improvement and refinements .

Author Response

We sincerely thank Reviewer 4 for their thoughtful and constructive feedback, which has substantially improved the manuscript’s clarity, organization, and scientific impact. 

Q1: The manuscript is rich in information, but the main storyline is sometimes diluted by excessive methodological detail. Please clarify the review’s central question and ensure each section explicitly contributes to answering it.

 A1: We appreciate this valuable feedback. We have clarified the review’s central question in the Introduction:

 “ The central question of this review is: “What known host cellular factors and mechanisms promote the LTR-RTE life cycle in plants?”.  

Each section has been revised to explicitly connect methodological discussions to this core question, with revised subsection introductions guiding readers on relevance to understanding host-LTR-RTE interactions.

Q2: Some subsections contain highly technical paragraphs that could be streamlined or reorganized. Adding clear transitional sentences between stages of the LTR-RTE life cycle will help readers follow the logic.

 A2: We agree. We have streamlined technical descriptions and added explicit transitional sentences throughout.

Q3: While comparisons with yeast, Drosophila, and mammals are informative, the plant-specific insights should be more emphasized. Several sections rely heavily on non-plant systems; consider guiding the reader more explicitly on how these findings translate to plants.

A3: Thank you for  suggestion. We have restructured each major section to prioritize plant systems first, then use comparative organisms to highlight mechanistic principles. We added explicit interpretation sentences (e.g., “In plants, …”) that translate findings from model systems to plant LTR-RTEs, making cross-kingdom insights more accessible. However, very limited information is available on plant host proteins.

Q4. Certain mechanistic descriptions (e.g., nuclear export, decapping, strand transfer) are excessively long for a review. Summarize where possible and highlight only what is necessary to build your argument. 

A4:  We have modified sections 2.2  and significantly condensed other section making more focus on plants and  by removing redundant mechanistic detail while retaining key concepts essential for understanding host factor roles.

Q5: Occasionally, speculative statements appear alongside confirmed data. Please distinguish hypotheses, assumptions, or unverified mechanisms clearly.

A5: We have systematically reviewed the manuscript and now clearly distinguish confirmed findings from speculative proposals using language such as “likely,” “may,” “remains to be determined,” and “we propose that.”

Q6: The review is comprehensive but largely descriptive. Consider incorporating critical evaluation of existing gaps, contradictions, and limitations in current plant LTR-RTE research. 

A6: We have strengthened the critical analysis throughout by explicitly identifying knowledge gaps (e.g., Section 2.2 on nuclear export, Section 3.4 on recombination), highlighting contradictions between plant and non-plant systems, and emphasizing limitations of current experimental approaches. A new “6. Concluding Remarks” paragraph has been added.

Q7. Figure 1 is informative but visually dense. Simplifying color coding or increasing text clarity could improve comprehension. 

A7: Figure 1 has been redesigned for improved clarity. We have simplified the color scheme (reduced from 6 to 4 distinct colors) and reorganized the figure legend.

Q8: The proposed research directions are excellent but could benefit from clearer prioritization—what approaches are immediately feasible for plants vs. long-term innovations?

 A8: We have significantly modified the Future Perspectives section.

Minor Points

Q9: Ensure consistency of terminology, especially regarding “host factors,” “LTR-RTE,” “retrotranscriptome,” and plant-specific nomenclature.

A9: We have conducted a comprehensive terminology audit throughout the manuscript. All key terms (“host factors,” “LTR-RTE,” “TE,” “retrotransposon RNA”) are now used consistently. 

Q10: Some sections would benefit from shorter sentences to aid readability.

A10: We have reviewed sentence structure throughout and reduced average sentence length.

Q11: A more concise abstract would increase its impact

A11: The abstract has been substantially shortened from approximately 250 words to 150 words while retaining the core message: LTR-RTEs depend on diverse host factors for mobility, plants employ unique silencing mechanisms, and emerging tools now enable functional dissection of these interactions..

Round 2

Reviewer 2 Report

Comments and Suggestions for Authors

The authors have thoroughly addressed the reviewers' comments, significantly improving the manuscript's clarity, depth, and analytical rigor. The revisions enhance the review's value as a resource for the plant transposon community. I recommend acceptance after minor revisions outlined below.

Key Strengths

Comprehensive Integration: The review effectively bridges knowledge gaps by comparing mechanisms across eukaryotes (plants, yeast, animals) and proposing plant-specific hypotheses.

Enhanced Figures: The expanded captions and improved figures (e.g., Figure 1 and 2) provide clearer visual summaries of complex processes.

Future-Oriented Perspectives: The addition of concrete experimental strategies (e.g., adapting CLEVR for plants, CRISPR screens) offers actionable guidance for future research.

Minor Revisions Required

Streamline Text: In Section 2.1, revise the sentence on chromatin remodelers for conciseness:

Suggested edit: "Unlike yeast Fun30 remodelers that control transcription start site selection, plant remodelers primarily modulate nucleosome occupancy to facilitate epigenetic silencing."

Correct Typographical Error: In Section 3.1, change "lariet" to "lariat."

Formatting Consistency: Ensure references strictly follow IJMS guidelines (e.g., uniform use of "et al.").

Conclusion

This review is a timely and valuable contribution to the field. The revisions have strengthened its scholarly impact, and the minor edits above will further polish the manuscript for publication.

Author Response

We thank the reviewer for the helpful suggestions. All corrections have been implemented as requested.
Q1: Streamline Text
Comment: In Section 2.1, revise the sentence on chromatin remodelers for conciseness.
Suggested edit: “Unlike yeast Fun30 remodelers that control transcription start site selection, plant remodelers primarily modulate nucleosome occupancy to facilitate epigenetic silencing.”
A1: Revised as suggested.
Q2: Correct Typographical Error
Comment: In Section 3.1, change “lariet” to “lariat.”
A2: Corrected.
Q3: Formatting Consistency
Comment: Ensure all references strictly follow IJMS guidelines (e.g., uniform use of “et al.”).
A3: Corrected accordingly.

Reviewer 4 Report

Comments and Suggestions for Authors

The revised manuscript presents a well-structured and timely review on host factors promoting the life cycle of LTR retrotransposons in plants. The topic is relevant and the scientific content is generally sound; however, a careful reading reveals that several editorial, typographical, terminological, and formatting issues still persist throughout the text. Although these issues do not undermine the scientific validity of the review, they should be addressed to improve clarity, consistency, and compliance with IJMS editorial standards.

First, there is inconsistent terminology and abbreviation usage across the manuscript. The terms “LTR-RTEs,” “LTR retrotransposons,” “LTR-RTs,” and “LTR retrotransposable elements” are used interchangeably. While all are scientifically valid, one primary term and abbreviation should be defined early and used consistently throughout the text, including figure legends. Similarly, capitalization is inconsistent (e.g., “Virus-like particles” vs “viral-like particles”), and a uniform lowercase style should be applied except at sentence beginnings.

In the Abstract and Introduction, minor grammatical and stylistic issues remain. For example, the phrase “Novel insertions arise in the final stages of the retrotransposon life cycle” would read more clearly as “arise during the final stages.” Additionally, some sentences show subject–verb agreement or clarity issues, such as “the life cycle of LTR-RTEs and their interaction with host factors and cellular compartments remain uncharacterized,” where “interactions” should be plural and “poorly characterized” would be more accurate. Redundant phrasing regarding dependence on host machinery is also present and could be streamlined.

Throughout Section 2, multiple typographical and encoding artifacts persist, including malformed hyphens in terms such as “LTR￾RTE,” which should be corrected uniformly. Gene and element names (e.g., ONSEN, EVADE) are italicized inconsistently, and standard gene nomenclature conventions should be applied consistently. In the nuclear export subsection, minor grammatical issues appear, such as incorrect singular/plural usage (“its mRNA” instead of “their mRNAs”), as well as ambiguous references to “this mechanism” without a clearly defined antecedent, which may confuse readers.

In the RNA processing and modification subsection, articles are occasionally missing (e.g., “Decapping is important RNA processing step” should read “an important RNA processing step”), and the notation for m⁶A methylation is inconsistent, alternating between “m6A” and “m⁶A.” The tense also shifts unnecessarily between past and present when describing established mechanisms; present tense should be used consistently for well-established knowledge.

In Section 3, which discusses translation, virus-like particle formation, and reverse transcription, several technical typographical errors are noticeable. Some protein names appear misspelled (e.g., “Dgfg10” instead of “Dfg10,” “Dpbb7” instead of “Dbp7”) and should be verified carefully against the cited literature. There is also redundancy in explaining the requirement for excess GAG over POL proteins, which is stated more than once within the same subsection. Terminology related to virus-like particles is again inconsistent, alternating between “viral-like” and “virus-like,” and should be standardized. Additionally, references to Figure 1 occasionally appear before the figure is fully introduced, which disrupts the logical flow.

In the reverse transcription and strand transfer sections, minor grammatical errors persist, such as incorrect plural forms (“this recombination events”) and subject–verb mismatches (“VLPs contains”). In the section on circular DNA formation, there is a notable encoding error where “cDNA” appears with a Cyrillic “с” instead of a Latin “c,” which should be corrected. Some sentences in this section are excessively long and would benefit from being split to improve readability and precision.

Finally, formatting and reference presentation require attention. Hyphenation of key terms such as “end-joining,” “end joining,” and “alt-EJ” is inconsistent, and citation formatting occasionally varies in spacing (e.g., “[7,8]” vs “[7, 8]”). These issues should be standardized according to journal style guidelines.

Comments on the Quality of English Language

The English language still needs additional improvement and refinements .

Author Response

We thank the reviewer for the helpful suggestions. All corrections have been implemented as requested.

Q1: First, there is inconsistent terminology and abbreviation usage across the manuscript. The terms “LTR-RTEs,” “LTR retrotransposons,” “LTR-RTs,” and “LTR retrotransposable elements” are used interchangeably. While all are scientifically valid, one primary term and abbreviation should be defined early and used consistently throughout the text, including figure legends. Similarly, capitalization is inconsistent (e.g., “Virus-like particles” vs “viral-like particles”), and a uniform lowercase style should be applied except at sentence beginnings.

A2: Corrected

Q2: In the Abstract and Introduction, minor grammatical and stylistic issues remain. For example, the phrase “Novel insertions arise in the final stages of the retrotransposon life cycle” would read more clearly as “arise during the final stages.” Additionally, some sentences show subject–verb agreement or clarity issues, such as “the life cycle of LTR-RTEs and their interaction with host factors and cellular compartments remain uncharacterized,” where “interactions” should be plural and “poorly characterized” would be more accurate. Redundant phrasing regarding dependence on host machinery is also present and could be streamlined.

A2: Corrected

Q3: Throughout Section 2, multiple typographical and encoding artifacts persist, including malformed hyphens in terms such as “LTR￾RTE,” which should be corrected uniformly. Gene and element names (e.g., ONSEN, EVADE) are italicized inconsistently, and standard gene nomenclature conventions should be applied consistently. In the nuclear export subsection, minor grammatical issues appear, such as incorrect singular/plural usage (“its mRNA” instead of “their mRNAs”), as well as ambiguous references to “this mechanism” without a clearly defined antecedent, which may confuse readers.

 A3: Corrected

Q4: In the RNA processing and modification subsection, articles are occasionally missing (e.g., “Decapping is important RNA processing step” should read “an important RNA processing step”), and the notation for m⁶A methylation is inconsistent, alternating between “m6A” and “m⁶A.” The tense also shifts unnecessarily between past and present when describing established mechanisms; present tense should be used consistently for well-established knowledge.

 A4: Corrected

Q5: In Section 3, which discusses translation, virus-like particle formation, and reverse transcription, several technical typographical errors are noticeable. Some protein names appear misspelled (e.g., “Dgfg10” instead of “Dfg10,” “Dpbb7” instead of “Dbp7”) and should be verified carefully against the cited literature. There is also redundancy in explaining the requirement for excess GAG over POL proteins, which is stated more than once within the same subsection. Terminology related to virus-like particles is again inconsistent, alternating between “viral-like” and “virus-like,” and should be standardized. Additionally, references to Figure 1 occasionally appear before the figure is fully introduced, which disrupts the logical flow.

A5: Corrected 

Q6: In the reverse transcription and strand transfer sections, minor grammatical errors persist, such as incorrect plural forms (“this recombination events”) and subject–verb mismatches (“VLPs contains”). In the section on circular DNA formation, there is a notable encoding error where “cDNA” appears with a Cyrillic “с” instead of a Latin “c,” which should be corrected. Some sentences in this section are excessively long and would benefit from being split to improve readability and precision.

 A6: Corrected 

Q7: Finally, formatting and reference presentation require attention. Hyphenation of key terms such as “end-joining,” “end joining,” and “alt-EJ” is inconsistent, and citation formatting occasionally varies in spacing (e.g., “[7,8]” vs “[7, 8]”). These issues should be standardized according to journal style guidelines.

A7: Corrected